# Retentive Network

## Abstract

In this work, we propose Retentive Network (RETNET) as a foundation architecture for large language models, simultaneously achieving training parallelism, low-cost inference, and good performance. We theoretically derive the connection between recurrence and attention. Then we propose the retention mechanism for sequence modeling, which supports three computation paradigms, i.e., parallel, recurrent, and chunkwise recurrent. Specifically, the parallel representation allows for training parallelism. The recurrent representation enables low-cost $O(1)$ inference, which improves decoding throughput, latency, and GPU memory without sacrificing performance. The chunkwise recurrent representation facilitates efficient long-sequence modeling with linear complexity, where each chunk is encoded parallelly while recurrently summarizing the chunks. Experimental results on language modeling show that RETNET achieves favorable scaling results, parallel training, low-cost deployment, and efficient inference.

## 1 Introduction

Transformer [51] has become the de facto architecture for large language models, which was initially proposed to overcome the sequential training issue of recurrent models [25]. However, training parallelism of Transformers is at the cost of inefficient inference, because of the $O(N)$ complexity per step and memory-bound key-value cache [42], which renders Transformers unfriendly to deployment. The growing sequence length increases GPU memory consumption as well as latency and reduces inference speed. Numerous efforts have continued to develop the next-generation architecture, aiming at retaining training parallelism and competitive performance as Transformers while having efficient $O(1)$ inference. It is challenging to achieve the above goals simultaneously.

There have been three main strands of research. First, linearized attention [27, 37] approximates standard attention scores $\exp(\boldsymbol{q} \cdot \boldsymbol{k})$ with kernels $\phi(\boldsymbol{q}) \cdot \phi(\boldsymbol{k})$, so that autoregressive inference can be rewritten in a recurrent form. However, the modeling capability and performance are worse than Transformers, which hinders the method's popularity. The second strand returns to recurrent models for efficient inference while sacrificing training parallelism. As a remedy, element-wise operators [36] are used for acceleration, however, representation capacity and performance are harmed. The third line explores replacing attention with other mechanisms, such as S4 [20], and its variants [11, 38]. None of the previous work can achieve strong performance and efficient inference at the same time compared to Transformers.

In this work, we propose retentive networks (RetNet), achieving low-cost inference, efficient long-sequence modeling, Transformer-comparable performance, and parallel model training simultaneously. Specifically, we introduce a multi-scale retention mechanism to substitute multi-head attention, which has three computation paradigms, i.e., parallel, recurrent, and chunkwise recurrent representations. First, the parallel representation empowers training parallelism to utilize GPU devices fully. Second, the recurrent representation enables efficient $O(1)$ inference in terms of memory and computation. The deployment cost and latency can be significantly reduced. Moreover, the implementation is greatly simplified without key-value cache tricks. Third, the chunkwise recurrent

representation can perform efficient long-sequence modeling. We parallelly encode each local block for computation speed while recurrently encoding the global blocks to save GPU memory.

We compare RetNet with Transformer and its variants. Experimental results on language modeling show that RetNet is consistently competitive in terms of both scaling curves and in-context learning. Moreover, the inference cost of RetNet is length-invariant. For a 7B model and 8k sequence length, RetNet decodes $8.4\times$ faster and saves 70% of memory than Transformers with key-value caches. During training, RetNet also achieves $3\times$ acceleration than standard Transformer with highly-optimized FlashAttention-2 [10]. Besides, RetNet's inference latency is insensitive to batch size, allowing enormous throughput. The intriguing properties make RetNet a potential candidate to replace Transformer for large language models.

## 2 Retentive Network

Retentive network (RetNet) is stacked with $L$ identical blocks, which follows a similar layout (i.e., residual connection, and pre-LayerNorm) as in Transformer [51]. Each RetNet block contains two modules: a multi-scale retention (MSR) module, and a feed-forward network (FFN) module. We introduce the MSR module in the following sections. Given an input sequence $x = x_1 \cdots x_{|x|}$, RetNet encodes the sequence in an autoregressive way. The input vectors $\{\boldsymbol{x}_i\}_{i=1}^{|x|}$ is first packed into $X^0 = [\boldsymbol{x}_1, \cdots, \boldsymbol{x}_{|x|}] \in \mathbb{R}^{|x| \times d_{\text{model}}}$, where $d_{\text{model}}$ is hidden dimension. Then we compute contextualized vector representations $X^l = \text{RetNet}_l(X^{l-1}), l \in [1, L]$.

### 2.1 Retention

In this section, we introduce the retention mechanism that has a dual form of recurrence and parallelism. So we can train the models in a parallel way while recurrently conducting inference.

Consider a sequence modeling problem that maps $v(n) \mapsto o(n)$ through states $\boldsymbol{s}_n$. Let $v_n, o_n$ denote $v(n), o(n)$ for simplicity. We formulate the mapping in a recurrent manner:

$$
\begin{aligned}
\boldsymbol{s}_n &= A\boldsymbol{s}_{n-1} + K_n^{\mathsf{T}} v_n, \quad A \in \mathbb{R}^{d \times d}, \quad K_n \in \mathbb{R}^{1 \times d} \\
o_n &= Q_n \boldsymbol{s}_n = \sum_{m=1}^{n} Q_n A^{n-m} K_m^{\mathsf{T}} v_m, \quad Q_n \in \mathbb{R}^{1 \times d}
\end{aligned}
\tag{1}
$$

where we map $v_n$ to the state vector $\boldsymbol{s}_n$, and then implement a linear transform to encode sequence information recurrently. Next, we make the projection $Q_n, K_n$ content-aware:

$$
Q = XW_Q, \quad K = XW_K
\tag{2}
$$

where $W_Q, W_K \in \mathbb{R}^{d \times d}$ are learnable matrices.

We diagonalize the matrix $A = \Lambda(\gamma e^{i\theta})\Lambda^{-1}$, where $\gamma, \theta \in \mathbb{R}^d$. Then we obtain $A^{n-m} = \Lambda(\gamma e^{i\theta})^{n-m}\Lambda^{-1}$. By absorbing $\Lambda$ into $W_Q$ and $W_K$, we can rewrite Equation (1) as:

$$
\begin{aligned}
o_n &= \sum_{m=1}^{n} Q_n (\gamma e^{i\theta})^{n-m} K_m^{\mathsf{T}} v_m \\
&= \sum_{m=1}^{n} (Q_n (\gamma e^{i\theta})^n)(K_m (\gamma e^{i\theta})^{-m})^{\mathsf{T}} v_m
\end{aligned}
\tag{3}
$$

where $Q_n (\gamma e^{i\theta})^n, K_m (\gamma e^{i\theta})^{-m}$ is known as xPos [45], i.e., a relative position embedding proposed for Transformer. We further simplify $\gamma$ as a scalar, Equation (3) becomes:

$$
o_n = \sum_{m=1}^{n} \gamma^{n-m} (Q_n e^{in\theta})(K_m e^{im\theta})^{\dagger} v_m
\tag{4}
$$

where $^{\dagger}$ is the conjugate transpose. The formulation is easily parallelizable within training instances.

In summary, we start with recurrent modeling as shown in Equation (1), and then derive its parallel formulation in Equation (4). We consider the original mapping $v(n) \mapsto o(n)$ as vectors and obtain the retention mechanism as follows.

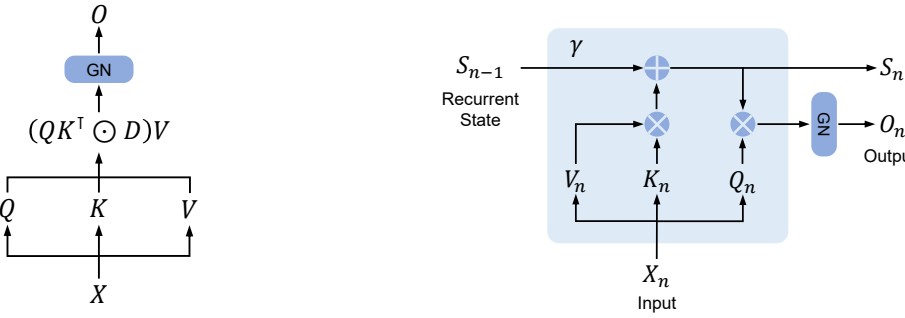

(a) Parallel representation.

(b) Recurrent representation.

Figure 1: RetNet has three equivalent computation paradigms, i.e., parallel, recurrent, and chunkwise recurrent representations. Given the same input, three paradigms obtain the same output. "GN" is short for GroupNorm.

**The Parallel Representation of Retention**  As shown in Figure 1a, the retention layer is defined as:

$$Q = (XW_Q) \odot \Theta, \quad K = (XW_K) \odot \overline{\Theta}, \quad V = XW_V$$

$$\Theta_n = e^{in\theta}, \quad D_{nm} = \begin{cases} \gamma^{n-m}, & n \geq m \\ 0, & n < m \end{cases} \tag{5}$$

$$\text{Retention}(X) = (QK^\mathsf{T} \odot D)V$$

where $D \in \mathbb{R}^{|x| \times |x|}$ combines causal masking and exponential decay along relative distance as one matrix, and $\overline{\Theta}$ is the complex conjugate of $\Theta$. In practice, we map $Q, K \in \mathbb{R}^d \to \mathbb{C}^{d/2}$, add the complex position embedding $\Theta$, then map them back to $\mathbb{R}^d$, following the implementation trick as in LLaMA [48, 44]. Similar to self-attention, the parallel representation enables us to train the models with GPUs efficiently.

**The Recurrent Representation of Retention**  As shown in Figure 1b, the proposed mechanism can also be written as recurrent neural networks (RNNs), which is favorable for inference. For the $n$-th timestep, we recurrently obtain the output as:

$$S_n = \gamma S_{n-1} + K_n^\mathsf{T} V_n$$

$$\text{Retention}(X_n) = Q_n S_n, \quad n = 1, \cdots, |x| \tag{6}$$

where $Q, K, V, \gamma$ are the same as in Equation (5).

**The Chunkwise Recurrent Representation of Retention**  A hybrid form of parallel representation and recurrent representation is available to accelerate training, especially for long sequences. We divide the input sequences into chunks. Within each chunk, we follow the parallel representation (Equation (5)) to conduct computation. In contrast, cross-chunk information is passed following the recurrent representation (Equation (6)). Specifically, let $B$ denote the chunk length. We compute the retention output of the $i$-th chunk via:

$$Q_{[i]} = Q_{Bi:B(i+1)}, \quad K_{[i]} = K_{Bi:B(i+1)}, \quad V_{[i]} = V_{Bi:B(i+1)}$$

$$R_i = K_{[i]}^\mathsf{T}(V_{[i]} \odot \zeta) + \gamma^B R_{i-1}, \quad \zeta_{ij} = \gamma^{B-i-1}$$

$$\text{Retention}(X_{[i]}) = \underbrace{(Q_{[i]} K_{[i]}^\mathsf{T} \odot D)V_{[i]}}_{\text{Inner-Chunk}} + \underbrace{(Q_{[i]} R_{i-1}) \odot \xi}_{\text{Cross-Chunk}}, \quad \xi_{ij} = \gamma^{i+1} \tag{7}$$

where $[i]$ indicates the $i$-th chunk, i.e., $x_{[i]} = [x_{(i-1)B+1}, \cdots, x_{iB}]$. The proof of the equivalence between recurrent representation and chunkwise recurrent representation is described in Appendix B.

## 2.2  Gated Multi-Scale Retention

We use $h = d_{\text{model}}/d$ retention heads in each layer, where $d$ is the head dimension. The heads use different parameter matrices $W_Q, W_K, W_V \in \mathbb{R}^{d \times d}$. Moreover, **m**ulti-**s**cale **r**etention (MSR) assigns

```
def ParallelRetention(
    q, k, v, # bsz * num_head * len * qkv_dim
    decay_mask): # num_head * len * len
    retention = q @ k.transpose(-1, -2)
    retention = retention * decay_mask
    output = retention @ v
    output = group_norm(output)
    return output

def RecurrentRetention(
    q, k, v, # bsz * num_head * qkv_dim
    past_kv, # bsz * num_head * qk_dim * v_dim
    decay): # num_head * 1 * 1
    current_kv = decay * past_kv + k.unsqueeze(-1) * v.
        unsqueeze(-2)
    output = torch.sum(q.unsqueeze(-1) * current_kv,
        dim=-2)
    output = group_norm(output)
    return output, current_kv
```

```
def ChunkwiseRetention(
    q, k, v, # bsz * num_head * chunk_size *
        qkv_dim
    past_kv, # bsz * num_head * qk_dim *
        v_dim
    decay_mask, # num_head * chunk_size *
        chunk_size
    chunk_decay, # num_head * 1 * 1
    inner_decay): # num_head * chunk_size
    retention = q @ k.transpose(-1, -2)
    retention = retention * decay_mask
    inner_retention = retention @ v
    cross_retention = (q @ past_kv) *
        inner_decay
    retention = inner_retention +
        cross_retention
    output = group_norm(retention)
    current_kv = chunk_decay * past_kv + k.
        transpose(-1, -2) @ v
    return output, current_kv
```

Figure 2: Pseudocode for the three computation paradigms of retention. Parallel implementation enables training parallelism to fully utilize GPUs. Recurrent paradigm enables low-cost inference. Chunkwise retention combines the above advantages (i.e., parallel within each chunk and recurrent across chunks), which has linear memory complexity for long sequences.

different $\gamma$ for each head. For simplicity, we set $\gamma$ identical among different layers and keep them fixed. In addition, we add a swish gate [23, 40] to increase the non-linearity of retention layers. Formally, given input $X$, we define the layer as:

$$
\begin{aligned}
\gamma &= 1 - 2^{-5-\operatorname{arange}(0,h)} \in \mathbb{R}^h \\
\text{head}_i &= \text{Retention}(X, \gamma_i) \\
Y &= \text{GroupNorm}_h(\text{Concat}(\text{head}_1, \cdots, \text{head}_h)) \\
\text{MSR}(X) &= (\text{swish}(XW_G) \odot Y)W_O
\end{aligned}
\tag{8}
$$

where $W_G, W_O \in \mathbb{R}^{d_{\text{model}} \times d_{\text{model}}}$ are learnable parameters, and GroupNorm [53] normalizes the output of each head, following SubLN proposed in [43]. Notice that the heads use multiple $\gamma$ scales, which results in different variance statistics. So we normalize the head outputs separately.

The pseudocode of retention is summarized in Figure 2.

**Retention Score Normalization** We utilize the scale-invariant nature of GroupNorm to improve the numerical precision of retention layers. Specifically, multiplying a scalar value within GroupNorm does not affect outputs and backward gradients, i.e., $\text{GroupNorm}(\alpha * \text{head}_i) = \text{GroupNorm}(\text{head}_i)$. We implement three normalization factors in Equation (5). First, we normalize $QK^\intercal$ as $QK^\intercal / \sqrt{d}$. Second, we replace $D$ with $\tilde{D}_{nm} = D_{nm} / \sqrt{\sum_{i=1}^n D_{ni}}$. Third, let $R$ denote the retention scores $R = QK^\intercal \odot D$, we normalize it as $\tilde{R}_{nm} = R_{nm} / \max(\sum_{i=1}^n |R_{ni}|, 1)$. Then the retention output becomes $\text{Retention}(X) = \tilde{R}V$. The above tricks do not affect the final results while stabilizing the numerical flow of both forward and backward passes, because of the scale-invariant property.

### 2.3 Overall Architecture of Retention Networks

For an $L$-layer retention network, we stack multi-scale retention (MSR) and feed-forward network (FFN) to build the model. Formally, the input sequence $\{x_i\}_{i=1}^{|x|}$ is transformed into vectors by a word embedding layer. We use the packed embeddings $X^0 = [\boldsymbol{x}_1, \cdots, \boldsymbol{x}_{|x|}] \in \mathbb{R}^{|x| \times d_{\text{model}}}$ as the input and compute the model output $X^L$:

$$
\begin{aligned}
Y^l &= \text{MSR}(\text{LN}(X^l)) + X^l \\
X^{l+1} &= \text{FFN}(\text{LN}(Y^l)) + Y^l
\end{aligned}
\tag{9}
$$

where $\text{LN}(\cdot)$ is LayerNorm [3]. The FFN part is computed as $\text{FFN}(X) = \text{gelu}(XW_1)W_2$, where $W_1, W_2$ are parameter matrices.

**Training** We use the parallel (Equation (5)) and chunkwise recurrent (Equation (7)) representations during the training process. The parallelization within sequences or chunks efficiently utilizes GPUs to accelerate computation. More favorably, chunkwise recurrence is especially useful for long-sequence training, which is efficient in terms of both FLOPs and memory consumption.

**Inference** The recurrent representation (Equation (6)) is employed during inference, which nicely fits autoregressive decoding. The $O(1)$ complexity reduces memory and inference latency while achieving equivalent results.

# 3 Experiments

We perform language modeling experiments to evaluate RetNet. First, we present the scaling curves of Transformer and RetNet. Second, we follow the training settings of StableLM-4E1T [50] to compare with open-source Transformer models in downstream benchmarks. Moreover, for training and inference, we compare speed, memory consumption, and latency. The training corpus is a curated compilation of The Pile [16], C4 [14], and The Stack [29].

## 3.1 Comparison with Transformer Variants

We compare RetNet with various efficient Transformer variants, including RWKV [36], H3 [11], Hyena [38], and Mamba [19]. We use LLaMA [48] architecture, including RMSNorm [59] and SwiGLU [40, 7] module, as the Transformer backbone, which shows better performance and stability. Consequently, other variants follow these settings. Specifically, Mamba does not have FFN layers so we only implement RMSNorm. For RetNet, the FFN intermediate dimension is $\frac{5}{3}d$ and the value dimensions in $W_G, W_V, W_O$ are also $\frac{5}{3}d$, where the overall parameters are still $12d^2$. All models have 400M parameters with 24 layers and a hidden dimension of 1024. For H3, we set the head dimension to 8. For RWKV, we use the TimeMix module to substitute self-attention layers while keeping FFN layers consistent with other models for fair comparisons. We train the models with 40k steps with a batch size of 0.25M tokens.

**Fine-Grained Language Modeling Evaluation** As shown in Table 1, we first report the language modeling perplexity of validation sets. Besides the overall validation set, following [2], we divide perplexity into "AR-Hit" and "First Occur". Specifically, AR-Hit contains the predicted tokens that are previously seen bigrams in the previous context, which evaluates the associative recall ability. "First Occur" has the predicted tokens that can not be recalled from the context. Among various Transformer variants, RetNet outperforms previous methods on both "AR-Hit" and "First Occur" splits, which is important for real-world use cases.

**Knowledge-Intensive Tasks** We also evaluate Massive Multitask Language Understanding (MMLU; [24]) answer perplexity to evaluate models on knowledge-intensive tasks. We report the average perplexity of the correct answers, i.e., given input [Question, ``Answer:'', Correct Answer], we calculate the perplexity of the "Correct Answer" part. RetNet achieves competitive results among the architectures.

| | Language Modeling | | | MMLU | | | | |
| | Valid. Set | AR-Hit | First-Occur | STEMs | Humanites | Social-Sci. | Others | Avg |
|---|---|---|---|---|---|---|---|---|
| Transformer [51] | 3.320 | 1.118 | 3.826 | 0.584 | 0.229 | 0.279 | 0.402 | 0.356 |
| *Transformer Variants* | | | | | | | | |
| Hyena [38] | 3.545 | 1.799 | 3.947 | 1.125 | 0.576 | 0.654 | 0.819 | 0.767 |
| RWKV [36] | 3.497 | 1.706 | 3.910 | 1.156 | 0.609 | 0.617 | 0.781 | 0.768 |
| Mamba [19] | 3.379 | 1.322 | 3.852 | 0.668 | 0.288 | 0.300 | 0.425 | 0.403 |
| H3 [11] | 3.563 | 1.722 | 3.986 | 1.169 | 0.532 | 0.637 | 0.792 | 0.752 |
| RetNet | **3.360** | **1.264** | **3.843** | **0.577** | **0.263** | **0.280** | **0.384** | **0.362** |

Table 1: Perplexity results on language modeling and MMLU [24] answers. We use the augmented Transformer architecture proposed in LLaMA [48] for reference. For language modeling, we report perplexity on both the overall validation set and fine-grained diagnosis sets [2], i.e., "AR-Hit" evaluates the associative recall capability, and "First-Occur" indicates the regular language modeling performance. Besides, we evaluate the answer perplexity of MMLU subsets.

## 3.2 Language Modeling Evaluation with Various Model Sizes

We train language models with various sizes (i.e., 1.3B, 2.7B, and 6.7B) from scratch. The training batch size is 4M tokens with 2048 maximal length. We train the models with 25k steps. The detailed hyper-parameters are described in Appendix E. We train the models with 512 AMD MI200 GPUs.

Figure 3 reports perplexity on the validation set for the language models based on Transformer and RetNet. We present the scaling curves with three model sizes, i.e., 1.3B, 2.7B, and 6.7B. RetNet achieves comparable results with Transformers. More importantly, the results indicate that RetNet is favorable in terms of size scaling. In addition to performance, RetNet training is quite stable in our experiments. Experimental results show that RetNet is a strong competitor to Transformer for large language models. Empirically, we find that RetNet starts to outperform Transformer when the model size is larger than 2B.

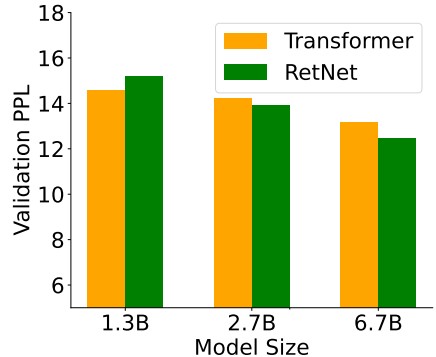

Figure 3: Validation perplexity (PPL) decreases along with scaling up the model size.

## 3.3 Long-Context Evaluation

We evaluate long-context modeling on the ZeroSCROLLS [41] benchmark. We train a hybrid model of size 2.7B, RetNet+, which stacks the attention and retention layers. Specifically, we insert one attention layer after every 3 retention layers. We follow most configurations of the 2.7B model as in Section 3.2. We scale the number of training tokens to 420B tokens. The batch size is 4M tokens. We first train the model with 4K length and then extend the sequence length to 16K for the last 50B training tokens. The rotation base scaling [55] is used for length extension.

Figure 4 reports the answer perplexity given various lengths of input document. It shows that both Transformer and RetNet+ perform better with longer input documents. The results indicate that the language models successfully utilize the long-distance context. Notice that the 12K and 16K results in Qasper are similar because the lengths of most documents are shorter than 16K. Moreover, RetNet+ obtains competitive results compared with Transformer for long-context modeling. Meanwhile, retention has better training and inference efficiency.

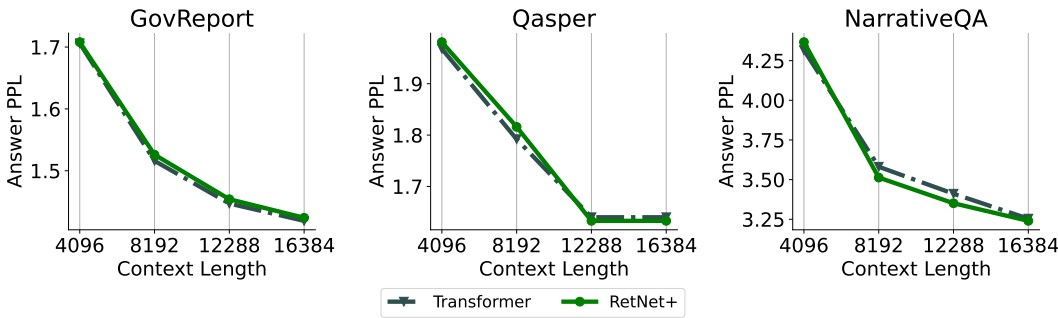

Figure 4: Answer perplexity decreases along with longer input documents. Transformer and RetNet+ obtain comparable performance for long-context modeling on the ZeroSCROLLS [41] benchmark.

## 3.4 Inference Cost

As shown in Figure 5, we compare memory cost, throughput, and latency of Transformer and RetNet during inference. Transformers reuse KV caches of previously decoded tokens. RetNet uses the recurrent representation as described in Equation (6). We evaluate the 6.7B model on the A100-80GB GPU. Figure 5 shows that RetNet outperforms Transformer in terms of inference cost.

**Memory** As shown in Figure 5a, the memory cost of Transformer increases linearly due to KV caches. In contrast, the memory consumption of RetNet remains consistent even for long sequences,

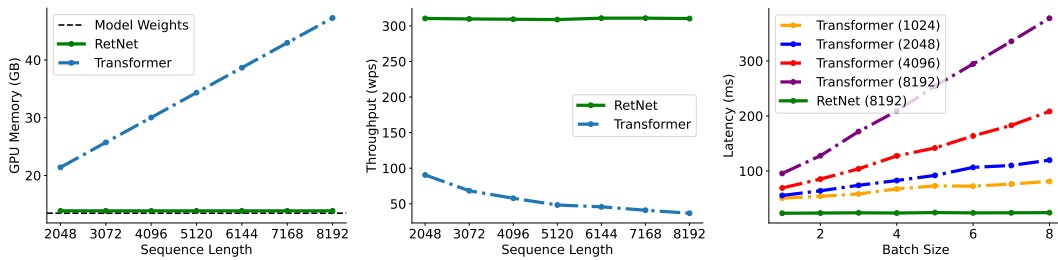

(a) GPU memory cost with varying sequence length.

(b) Inference throughput with varying sequence length.

(c) Inference latency with different batch sizes.

Figure 5: Inference cost of Transformer and RetNet with a model size of 6.7B. RetNet outperforms Transformers in terms of memory consumption, throughput, and latency.

requiring much less GPU memory to host RetNet. The additional memory consumption of RetNet is almost negligible (i.e., about 3%) while the model weights occupy 97%.

**Throughput**   As presented in Figure 5b, the throughput of Transformer drops along with the decoding length increases. In comparison, RetNet has higher and length-invariant throughput during decoding, by utilizing the recurrent representation of retention.

**Latency**   Latency is an important metric in deployment that greatly affects the user experience. We report the decoding latency in Figure 5c. Experimental results show that increasing batch size renders the Transformer's latency larger. Moreover, the latency of Transformers grows faster with longer input. In order to make latency acceptable, we have to restrict the batch size, which harms the overall inference throughput of Transformers. By contrast, RetNet's decoding latency outperforms Transformers and stays almost the same across different batch sizes and input lengths.

### 3.5   Training Throughput

Figure 6 compares the training throughput of Transformer and RetNet, where the training sequence lengths range from 8192 to 65536. The model size is 3.5B, where the hidden dimension is 3072 and the layer size is 28. We use highly optimized FlashAttention-2 [10] for Transformers. In comparison, we implement chunk recurrent representation (Equation (7)) using Triton [46], where the computation is both memory-friendly and computationally efficient. The chunk size is set to 256. We evaluate the results with eight Nvidia H100-80GB GPUs because FlashAttention-2 is highly optimized for H100 cards.

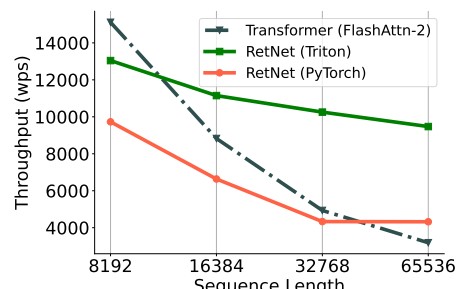

Figure 6: Training throughput (word per second; wps) of Transformer with FlashAttention-2 [10] and RetNet.

Experimental results show that RetNet has higher training throughput than Transformers. The acceleration ratio increases as the sequence length is longer. When the training length is 64k, RetNet's throughput is about 3 times than Transformer's.

### 3.6   Zero-Shot and Few-Shot Evaluation on Downstream Tasks

We also compare the language models on a wide range of downstream tasks. We evaluate zero-shot and 4-shot learning with the 6.7B models. As shown in Table 2, the datasets include HellaSwag (HS; [57]), BoolQ [8], COPA [52], PIQA [6], Winograd, Winogrande [30], and StoryCloze (SC; [34]). The accuracy numbers are consistent with language modeling perplexity presented in Figure 3. RetNet achieves comparable performance with Transformer on zero-shot and in-context learning settings.

### 3.7   Ablation Studies

We ablate various design choices of RetNet and report the language modeling results in Table 3. The evaluation settings and metrics are the same as in Section 3.1.

|          | HS   | BoolQ | COPA | PIQA | Winograd | Winogrande | SC   | Avg   |
|----------|------|-------|------|------|----------|------------|------|-------|
| *Zero-Shot Performance* | | | | | | | | |
| Transformer | 55.9 | 62.0 | 69.0 | 74.6 | 69.5 | 56.5 | 75.0 | 66.07 |
| RetNet | **60.7** | **62.2** | **77.0** | **75.4** | **77.2** | **58.1** | **76.0** | **69.51** |
| *Few-shot Performance (4-Shot)* | | | | | | | | |
| Transformer | 55.8 | 58.7 | 71.0 | 75.0 | 71.9 | 57.3 | 75.4 | 66.44 |
| RetNet | **60.5** | **60.1** | **78.0** | **76.0** | **77.9** | **59.9** | **75.9** | **69.76** |

Table 2: Zero-shot and few-shot learning performance. The language model size is 6.7B.

**Architecture**  We ablate the swish gate and GroupNorm as described in Equation (8). Table 3 shows that the above two components improve performance. First, the gating module is essential for enhancing non-linearity and improving model capability. Notice that we use the same parameter allocation as in Transformers after removing the gate. Second, group normalization in retention balances the variances of multi-head outputs, which improves training stability and language modeling results.

**Multi-Scale Decay**  Equation (8) shows that we use different $\gamma$ as the decay rates for the retention heads. In the ablation studies, we examine removing $\gamma$ decay (i.e., "$- \gamma$ decay") and applying the same decay rate across heads (i.e., "$-$ multi-scale decay"). Specifically, ablating $\gamma$ decay is equivalent to $\gamma = 1$. In the second setting, we set $\gamma = 1 - 2^{-6.5}$ for all heads. Table 3 indicates that both the decay mechanism and using multiple decay rates can improve the language modeling performance.

**Head Dimension**  As indicated by the recurrent perspective of Equation (1), the head dimension implies the memory capacity of hidden states. In ablation, we reduce the default head dimension from 256 to 64, i.e., 64 for queries and keys, and $\lfloor \frac{5}{3} \times 64 \rfloor \approx 108$ for values. We keep the hidden dimension $d_{\text{model}}$ the same. Accordingly, we adjust the multi-scale decay as $\gamma = 1 - 2^{-5-arange(0,h)/4}$ to keep the same decay range. Table 3 shows that the larger head dimension achieves better performance.

|          | Language Modeling | | | MMLU | | | | |
|----------|-----------|--------|-------------|-------|-----------|-------------|--------|-------|
|          | Valid. Set | AR-Hit | First-Occur | STEMs | Humanites | Social-Sci. | Others | Avg   |
| RetNet | **3.360** | **1.264** | **3.843** | **0.577** | **0.263** | **0.280** | **0.384** | **0.362** |
| $-$ swish gate | 3.509 | 1.366 | 4.002 | 0.599 | 0.285 | 0.315 | 0.421 | 0.390 |
| $-$ GroupNorm | 3.367 | 1.302 | 3.843 | 0.630 | 0.295 | 0.327 | 0.438 | 0.406 |
| $- \gamma$ decay | 3.920 | 2.122 | 4.334 | 0.958 | 0.566 | 0.571 | 0.694 | 0.681 |
| $-$ multi-scale decay | 3.524 | 1.768 | 3.928 | 0.921 | 0.433 | 0.471 | 0.590 | 0.582 |
| Reduce head dim. | 3.397 | 1.331 | 3.872 | 0.637 | 0.272 | 0.294 | 0.393 | 0.384 |

Table 3: Perplexity results on language modeling and MMLU [24] answers. For language modeling, we report perplexity on both the overall validation set and fine-grained diagnosis sets [2], i.e., "AR-Hit" evaluates the associative recall capability, and "First-Occur" indicates the regular language modeling performance. Besides, we evaluate the answer perplexity of the MMLU subsets.

## 3.8  Results on Vision Tasks

We also compare RetNet with vision Transformers [15, 47] in Table 4, where bidirectional encoders are evaluated. Unlike causal language models, the vision encoders do not require recurrent representations. Specifically, we use retention as follows:

$$Q = (XW_Q) \odot \Theta, \quad K = (XW_K) \odot \overline{\Theta}, \quad V = XW_V$$
$$\text{Retention}(X) = (QK^\intercal)V = Q(K^\intercal V)$$

where multi-scale decay is removed in bidirectional computation. Notice that we can compute retention in different orders. Similar to linear attention [27], the $Q(K^\intercal V)$ paradigm is an efficient operator in bidirectional settings, especially for high-resolution images.

We perform experiments on ImageNet-1K classification [13], COCO object detection [32], and ADE20K semantic segmentation [60]. We compare RetNet with DeiT [47] which is a well-tuned vision Transformer. Besides, we follow [21] and plug in a depth-wise convolution in experiments. We adopt the DeiT-M size, which has about 38M parameters. For ImageNet-1K image classification,

| | ImageNet | COCO | | | ADE20K | |
|---|---|---|---|---|---|---|
| | Acc | $AP^b$ | $AP^b_{50}$ | $AP^b_{75}$ | mIoU | mAcc |
| DeiT [47] | 80.76 | 0.458 | 0.678 | 0.502 | 43.52 | 55.08 |
| RetNet | 81.57 | 0.457 | 0.669 | 0.488 | 44.13 | 56.12 |

Table 4: Results on vision tasks, i.e., image classification (ImageNet), object detection (COCO), and semantic segmentation (ADE20K). RetNet achieves competitive performance with DeiT, which is a well-tuned vision Transformer.

we use AdamW [33] for 300 epochs, and 20 epochs of linear warm-up. The learning rate is $1 \times 10^{-3}$, the batch size is 1024, and the weight decay is 0.05. For COCO object detection, we use Mask R-CNN [22] as the task head, and the above models pre-trained on ImageNet as the backbone with 3x schedules. In ADE20K experiments, we use UperNet [54] as the segmentation head. The detailed configuration can be found in Appendix H.

Table 4 shows the results across various vision tasks. RetNet is competitive compared with DeiT. For classification and segmentation, RetNet is slightly better than DeiT, where RetNet achieves 0.81% accuracy improvement on ImageNet and 0.61% mIoU improvement on ADE20K. For object detection, the results are comparable.

## 4 Related Work

Numerous efforts are focused on reducing the quadratic complexity of attention mechanisms. Linear attention [27] uses various kernels $\phi(q_i)\phi(k_j)/\sum_{n=1}^{|x|} \phi(q_i)\phi(k_n)$ to replace the $\mathrm{softmax}$ function. In contrast, we reexamine sequence modeling from scratch, rather than aiming at approximating $\mathrm{softmax}$. AFT [58] simplifies dot-product attention to element-wise and moves $\mathrm{softmax}$ to key vectors. RWKV [36] replaces AFT's position embeddings with exponential decay and runs the models recurrently for training and inference. In comparison, retention preserves high-dimensional states to encode sequence information, which contributes to expressive ability and better performance. S4 [20] unifies convolution and recurrence format and achieves $O(N \log N)$ training complexity leveraging the FFT kernel. Unlike Equation (2), if $Q_n$ and $K_n$ are content-unaware, the formulation can be degenerated to S4 [20]. Hyena [38] generates the convolution kernels, achieving sub-quadratic training efficiency but keeping $O(N)$ complexity in single-step inference. Recently, most related work has focused on modifying $\gamma$ in Equation (6) as a data-dependent variable, such as Mamba [19], GLA [56], Gateloop [28], and xLSTM [4]. Another strand explores hybrid architectures [31, 12] that interleave the above components with attention layers.

In addition, we discuss the training and inference efficiency of some related methods. Let $D$ denote the hidden dimension, $H$ the head dimension, and $N$ the sequence length. For training, RWKV's token-mixing complexity is $O(DN)$, and Mamba's complexity is $O(DHN)$ with optimized CUDA kernels. Hyena's is $O(DN \log N)$ with Fast Fourier Transform acceleration. In comparison, the chunk-wise recurrent representation is $O(DN(B + H))$, where $B$ is the chunk size, and we usually set $H = 256, B \leq 512$. However, chunk-wise computation is highly parallelized, enabling efficient hardware usage. For large model size (i.e., larger $D$) or sequence length, the additional $b + h$ has negligible effects. For inference, among the efficient architectures compared, Hyena has the same complexity (i.e., $O(N)$ per step) as Transformer, while the others can perform $O(1)$ decoding.

## 5 Conclusion

We propose retentive networks (RetNet) for sequence modeling, which enables various representations, i.e., parallel, recurrent, and chunkwise recurrent. RetNet achieves significantly better inference efficiency (in terms of memory, speed, and latency), favorable training parallelization, and competitive performance compared with Transformers. The above advantages make RetNet an ideal successor to Transformers for large language models, especially considering the deployment benefits brought by the $O(1)$ inference complexity. In the future, we are interested in deploying RetNet on various edge devices, such as mobile phones.

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

# A  Scaling Up Number of Training Tokens

We scale up the number of training tokens to 350B for the 3B-size models. We compare with strong Transformer checkpoints including OpenLLaMA [18] and StableLM [50]. Moreover, we reproduce a Transformer language model (named Transformer$_{\text{Repro}}$) for apple-to-apple comparison.

Our model RetNet+ follows the same configuration as in Section 3.3, which is a hybrid model. The model's hidden size is 3072, and the number of layers is 28. Without vocabulary embedding, the total number of parameters is 3.17B, which is between StableLM-3B-4E1T (2.7B) and OpenLLaMA-3B-v1 (3.19B). The batch size is 4M tokens. The training length is 4k. The learning rate is $3.2 \times 10^{-4}$ with 1000 warm-up steps and linear learning rate decay. The training corpus includes The Pile [16] and RedPajama [9]. Transformer$_{\text{Repro}}$ follows the exact same setting.

Table 5 reports accuracy numbers on the Harness-Eval benchmark [17]. We directly follow the evaluation protocol. The results show that RetNet+ achieves a performance comparable to Transformer$_{\text{Repro}}$ on language tasks. Notice that OpenLLaMA-3B-v1 and StableLM-3B use different learning rate schedules. The results of these two models are used for reference purposes.

| Model | ARC-C | ARC-C$_{\text{norm}}$ | ARC-E | ARC-E$_{\text{norm}}$ | Hellaswag | Hellaswag$_{\text{norm}}$ |
|---|---|---|---|---|---|---|
| OpenLLaMA-3B-v1 | 0.303 | 0.323 | 0.641 | 0.599 | 0.449 | 0.608 |
| StableLM-3B | — | — | 0.649 | 0.610 | — | — |
| Transformer$_{\text{Repro}}$ | **0.322** | **0.354** | 0.668 | **0.633** | 0.476 | 0.633 |
| RetNet+ | 0.321 | 0.347 | **0.675** | 0.613 | **0.478** | **0.639** |

| Model | OBQA | OBQA$_{\text{norm}}$ | PIQA | PIQA$_{\text{norm}}$ | Winogrande | Avg |
|---|---|---|---|---|---|---|
| OpenLLaMA-3B-v1 | 0.222 | 0.348 | 0.713 | 0.724 | 0.594 | 0.502 |
| StableLM-3B | — | — | **0.759** | **0.763** | 0.608 | — |
| Transformer$_{\text{Repro}}$ | **0.258** | 0.358 | 0.746 | 0.755 | 0.612 | **0.529** |
| RetNet+ | **0.258** | **0.362** | 0.750 | **0.763** | **0.614** | **0.529** |

Table 5: Accuracy on the Harness-Eval benchmark. All models are trained with 350B tokens with a batch size of 4M tokens. The results of OpenLLaMA-3B-v1 are taken from their official repository (https://bit.ly/openllama-350b-results), and StableLM-3B from their technical report (https://bit.ly/StableLM-3B-4E1T).

# B  Equivalence Between Chunk-wise Recurrent Representation and Recurrent Representation

We illustrate the equivalence between the recurrent representation and the chunk-wise recurrent representation. Specifically, let $B$ denote the chunk length. For the output $O_n$, $n$ can be divided as $n = kB + r$ where $B$ is the chunk size. Following Equation 6, we have:

$$
\begin{aligned}
O_n &= \sum_{m=1}^{n} \gamma^{n-m} Q_n K_m^\mathsf{T} V_m \\
&= (Q_n K_{kB+1:n}^\mathsf{T} \odot \Gamma) V_{kB+1:n} + (Q_n \gamma^r) \sum_{c=0}^{k-1} \sum_{m=1}^{B} (K_{m+cB}^\mathsf{T} V_{m+cB} \gamma^{B-m}) \gamma^{(k-1-c)B} \\
&= (Q_n K_{kB+1:n}^\mathsf{T} \odot \Gamma) V_{kB+1:n} + (Q_n \gamma^r) \sum_{c=1}^{k} (K_{[c]}^\mathsf{T} (V_{[c]} \odot \zeta)) \gamma^{(k-c)B} \\
&= (Q_n K_{kB+1:n}^\mathsf{T} \odot \Gamma) V_{kB+1:n} + (Q_n \gamma^r) R_{i-1}
\end{aligned}
\tag{10}
$$

where $\Gamma_i = \gamma^{n-i}$, $\zeta_{ij} = \gamma^{B-m}$, and $[i]$ indicates the $i$-th chunk, i.e., $x_{[i]} = [x_{(i-1)B+1}, \cdots, x_{iB}]$. Then we write $R_n$ as a recurrent function and compute the retention output of the $i$-th chunk via:

$$
\begin{aligned}
R_i &= K_{[i]}^\mathsf{T} (V_{[i]} \odot \zeta) + \gamma^B R_{i-1} \\
\zeta_{ij} &= \gamma^{B-i}, \quad \xi_{ij} = \gamma^i \\
\text{Retention}(X_{[i]}) &= \underbrace{(Q_{[i]} K_{[i]}^\mathsf{T} \odot D) V_{[i]}}_{\text{Inner-Chunk}} + \underbrace{(Q_{[i]} \odot \xi) R_{i-1}}_{\text{Cross-Chunk}}
\end{aligned}
\tag{11}
$$

Finally, we show that the chunkwise recurrent representation is equivalent to the other representations.

## C  Results with Different Context Lengths

As shown in Table 6, we report the results of language modeling with different context lengths. In order to make the numbers comparable, we use 2048 text chunks as evaluation data and only compute the perplexity for the last 128 tokens. Experimental results show that RetNet performs comparably with Transformer in different context lengths.

| Model | 512 | 1024 | 2048 |
|---|---|---|---|
| Transformer | 13.55 | 12.56 | 12.35 |
| RetNet | 13.09 | 12.14 | 11.98 |

Table 6: Language modeling perplexity of RetNet and Transformer with different context length. The results show that RetNet has a consistent advantage across sequence length.

## D  Hyperparameters Used in Section 3.1

We use LLaMA [48] architecture, including RMSNorm [59] and SwiGLU [40, 7] module, as the Transformer backbone, which shows better performance and stability. The weights of word embedding and $\mathrm{softmax}$ projection are shared. Consequently, other variants follow these settings. For RetNet, the FFN intermediate dimension is $\frac{5}{3}d$ and the value dimensions in $W_G, W_V, W_O$ are also $\frac{5}{3}d$, where the overall parameters are still $12d^2$.

For H3, we set the head dimension to 8. For RWKV, we use the TimeMix module to substitute self-attention layers while keeping FFN layers consistent with other models for fair comparisons. For Mamba, we follow all the details in the paper [19], where double-SSM layers are implemented instead of "SSM + SwiGLU". In addition to RetNet and Mamba, the FFN intermediate dimension is all $\frac{8}{3}d$. All models have 400M parameters, 24 layers, and a hidden dimension of 1024. We train the models with 40k steps and a batch size of 0.25M tokens.

| Params | Values |
|---|---|
| Layers | 24 |
| Hidden size | 1024 |
| Vocab size | 100,288 |
| Heads | 24 |
| Adam $\beta$ | (0.9, 0.98) |
| LR | $1.5 \times 10^{-4}$ |
| Batch size | 0.25M |
| Warmup steps | 375 |
| Weight decay | 0.05 |
| Dropout | 0.0 |

Table 7: Hyperparamters used for the architecture comparison in Section 3.1.

## E  Hyperparameters Used in Section 3.2

We re-allocate the parameters in MSR and FFN for fair comparisons. Let $d$ denote $d_{\mathrm{model}}$ for simplicity here. In Transformers, there are about $4d^2$ parameters in self-attention where $W_Q, W_K, W_V, W_O \in \mathbb{R}^{d \times d}$, and $8d^2$ parameters in FFN where the intermediate dimension is $4d$. In comparison, RetNet has $8d^2$ parameters in retention, where $W_Q, W_K \in \mathbb{R}^{d \times d}, W_G, W_V \in \mathbb{R}^{d \times 2d}, W_O \in \mathbb{R}^{2d \times d}$. Notice that the head dimension of $V$ is twice $Q, K$, similar to GAU [26]. The widened dimension is projected back to $d$ by $W_O$. In order to keep the parameter number the same as Transformer, the FFN intermediate dimension in RetNet is $2d$. Meanwhile, we set the head dimension to 256, i.e., 256 for

queries and keys, and 512 for values. For fair comparison, we keep $\gamma$ identical among different model sizes, where $\gamma = 1 - e^{\text{linspace}(\log 1/32, \log 1/512, h)} \in \mathbb{R}^h$ instead of the default value in Equation (8).

| Hyperparameters | 1.3B | 2.7B | 6.7B |
|---|---|---|---|
| Layers | 24 | 32 | 32 |
| Hidden size | 2048 | 2560 | 4096 |
| FFN size | 4096 | 5120 | 8192 |
| Heads | 8 | 10 | 16 |
| Learning rate | $6 \times 10^{-4}$ | $3 \times 10^{-4}$ | $3 \times 10^{-4}$ |
| LR scheduler | | Linear decay | |
| Warm-up steps | | 375 | |
| Tokens per batch | | 4M | |
| Adam $\beta$ | | (0.9, 0.98) | |
| Training steps | | 25,000 | |
| Gradient clipping | | 2.0 | |
| Dropout | | 0.1 | |
| Weight decay | | 0.05 | |

Table 8: Hyperparamters used for language modeling in Section 3.2.

## F    Results on Open-Ended Generation Tasks

Table 9 presents one-shot performance on two open-ended question-answering tasks, including SQUAD [39] and WebQS [5], with 6.7B models as follows. We report the recall metric in the table, i.e., whether the answers are contained in the generated response.

| Dataset | SQUAD | WebQS |
|---|---|---|
| Transformer | 67.7 | 36.4 |
| RetNet | 72.7 | 40.4 |

Table 9: Answer recall of RetNet and Transformer on open-ended question answering.

## G    Inference Cost of Grouped-Query Retention

We compare with grouped-query attention [1] and evaluate the method in the context of RetNet. Grouped-query attention makes a trade-off between performance and efficiency, which has been successfully verified in LLaMA2 34B/70B [49]. The method reduces the overhead of key/value cache during inference. Moreover, the performance of grouped-query attention is better than multi-query attention [42], overcoming the quality degradation brought by using one-head key value.

As shown in Table 10, we compare the inference cost with grouped-query attention and apply the method for RetNet. For the LLaMA2 70B model, the number of key/value heads is reduced by $8\times$, where the query head number is 64 while the key/value head number is 8. For RetNet-70B, the parameter allocation is identical to LLaMA [48], where the dimension is 8192, and the head number is 32 for RetNet. For RetNet-70B-GQ2, the key-value head number is 16, where grouped-query retention is applied. We run the inference with four A100 GPUs without quantization.

When the batch size is 256, LLaMA2 runs out of memory while RetNet without group query still has a high throughput. When equipped with grouped-query retention, RetNet-70B achieves 38% acceleration and saves 30% memory.

We evaluate LLaMA2 under 2k and 8k lengths separately. The batch size is reduced to 8 so that LLaMA2 can run without out of memory. Table 10 shows that the inference cost of Transformers increases with the sequence length. In contrast, RetNet is length-invariant. Moreover, RetNet-70B-GQ2 achieves better latency, throughput, and GPU memory than LLaMA2-70B-2k/8k equipped

with grouped-query attention. Notice that the evaluation metrics are averaged over positions of different sequence lengths for a fair comparison, rather than only considering the inference cost of the maximum length.

| Model | Batch Size | Latency (ms)↓ | Throughput (wps)↑ | Memory (GB)↓ |
|---|---|---|---|---|
| LLaMA2-70B-2k | 256 | — | — | OOM |
| LLaMA2-70B-8k | 256 | — | — | OOM |
| RetNet-70B | 256 | 639.1 | 410.19 | 72.469 |
| RetNet-70B-GQ2 | 256 | 461.8 | 567.66 | 52.726 |
| LLaMA2-70B-2k | 8 | 184.5 | 44.42 | 33.374 |
| LLaMA2-70B-8k | 8 | 277.7 | 29.50 | 37.386 |
| RetNet-70B-GQ2 | 8 | 106.2 | 77.02 | 32.301 |

Table 10: Inference cost of RetNet and LLaMA2-70B with difference batch size and length. LLaMA2-70B is equipped with grouped-query attention, reducing key/value heads by 8×. "-GQ2" means grouped-query retention, which reduces half of key/value heads. "-2k" and "-8k" indicate sequence length for LLaMA2, while RetNet is length-invariant. RetNet is capable of large-batch inference and is favourable in terms of latency, throughput, and GPU memory.

# H  Hyperparameters Used in Section 3.8

| Hyperparameters | DeiT | RetNet |
|---|---|---|
| Layers | 12 | 12 |
| Hidden size | 512 | 512 |
| Patch size | 16 | 16 |
| FFN size | 2048 | 1024 |
| Heads | 8 | 2 |
| Learning rate | $1 \times 10^{-3}$ | |
| LR scheduler | Cosine decay | |
| Batch size | 1024 | |
| Epochs | 300 | |
| Warmup epochs | 5 | |
| Smoothing | 0.1 | |
| Weight decay | 0.05 | |
| Drop path | 0.3 | |

Table 11: Hyperparamters used for the ImageNet experiments in Section 3.8.

