# OpenReview forum: "Retentive Network"
_NeurIPS.cc/2024/Conference — Submitted to NeurIPS 2024_

### Official Review · Reviewer_ZaX6 · 2024-07-12

**Soundness:** 3
**Presentation:** 3
**Contribution:** 3
**Rating:** 5
**Confidence:** 4

**Summary:**

The paper proposes the Retentive Network (RetNet) as a foundation architecture for large language models. RetNet has a multi-scale retention mechanism with three computation paradigms: parallel, recurrent, and chunkwise recurrent.
The retention mechanism starts with a recurrent modeling formulation and derives a parallel formulation. It maps input vectors to state vectors recurrently and implements a linear transform to encode sequence information. Then, it makes the projection content-aware by using learnable matrices. The retention layer is defined using these matrices and a complex position embedding, combining causal masking and exponential decay along relative distance.
It achieves low-cost inference, efficient long-sequence modeling, comparable performance to Transformers, and parallel training. Experimental results show its superiority in language modeling, inference cost, and training throughput.

**Strengths:**

1. The RetNet also shows competitive performance in language modeling and knowledge-intensive tasks compared to other Transformer variants and has the potential to replace Transformers for large language models.
2. Achieves significantly better inference efficiency in terms of memory, speed, and latency.

**Weaknesses:**

1. The paper presents the scaling curves of RetNet and Transformer with model sizes ranging from 1.3B to 6.7B, concluding that RetNet is favorable in terms of size scaling and starts to outperform Transformer when the model size is larger than 2B. However, it does not provide a detailed explanation for this trend. Understanding the underlying reasons for this performance difference with increasing model size could provide more insights into the effectiveness of RetNet and its potential advantages over Transformer.
2. The use of $\gamma$ in the RetNet may appear somewhat heuristic. The paper assigns different $\gamma$ for each head in the multi-scale retention (MSR) module and keeps them fixed among different layers. While this approach is used to achieve certain effects, such as enhancing the non-linearity of the retention layers and improving the model's performance, the specific rationale for choosing these values and the potential impact on the model's behavior could be further explained.

**Questions:**

1. Can you provide more insights into why RetNet starts to outperform Transformer when the model size is larger than 2B? What specific characteristics or mechanisms of RetNet contribute to this improved performance at larger scales? Have you tried different learning rates to investigate their impact on the scaling of the RetNet model? If so, what were the results and how did they affect the model's performance and scalability?
2. A more detailed discussion on the selection of $\gamma$ and its effects on the model's performance, as well as how it compares to other possible approaches or values, would be beneficial in providing a deeper understanding of the RetNet's functionality. Additionally, exploring the sensitivity of the model to different values of $\gamma$ or conducting experiments to justify the choice could strengthen the argument for using this particular heuristic.

---

> ### Author Rebuttal · Authors · 2024-08-07
>
> Thanks for your positive comments.
>
>
> >Q1: Can you provide more insights into why RetNet starts to outperform Transformer when the model size is larger than 2B?
>
> A1: Because of the recurrence nature of the proposed method, the dimension of "hidden states" is critical for model performance, which is similar to the concept of LSTM hidden size. We find that expanding the width is beneficial for retention, especially for smaller size. Along with model size increases, the hidden size becomes larger.
>
>
> ---
> >Q2: Have you tried different learning rates to investigate their impact on the scaling of the RetNet model? If so, what were the results and how did they affect the model's performance and scalability?
>
> A2: We follow the learning rate schedule used by Transformers for fair comparisons in the paper. The suggested investigation would be valuable to obtain the relation between optimal learning rate and model size.
>
>
> ---
> >Q3: Use of $\gamma$? How is $\gamma$ determined?
>
> A3: As descibed in Line 66 and Eq. (3), the decay term $\gamma$ is introduced by diagonalizing the matrix $A$. As shown in Table 3, the ablation "without $\gamma$ decay" performs worse than the proposed retention, indicating the effectiveness of our design. The physical meaning of $\gamma$ is relative position bias, such as Alibi[1], and xPos[2]. Following the above works, we assign different decay values for the heads. Moreover, the decay speed is similar with previous position embedding works, rather than heuristic search of $\gamma$.
>
> [1] Train Short, Test Long: Attention with Linear Biases Enables Input Length Extrapolation
>
> [2] A Length-Extrapolatable Transformer
>
> We hope the above explanation clarifies the rationale behind our designs.

---

> > ### Comment · Reviewer_ZaX6 · 2024-08-13
> >
> > Thank you for your reply. I decided to maintain my original score.

---

### Official Review · Reviewer_maEf · 2024-07-13

**Soundness:** 2
**Presentation:** 2
**Contribution:** 2
**Rating:** 4
**Confidence:** 5

**Summary:**

The authors propose a linear attention model called RetNet for language modeling, which has a linear training complexity and constant inference complexity.

**Strengths:**

1. RetNet has both linear time complexity and constant inference memory complexity.
2. RetNet has a chunk recurrent form which can be beneficial for speculative decoding.

**Weaknesses:**

1. The authors introduce a new term called "Retention," but this is essentially the same as Linear Attention without the denominator, which has already been proposed in [1].
2. Lack of comparison with the baselines on open source pretraining data. All the training experiments are conducted on in-house data mixtures, which harms the reproducibility.
3. The paper doesn't compare RetNet with other linear attention model (such as GLA, RWKV, Mamba) on downstream tasks with standard metrics instead of perplexity. Table 2 only include RetNet and Transformer. The efficiency measurment of RetNet+ is absent.
4. The evaluation on MMLU/Qasper is using perplexity but not the widely-used accuracy/F1 metric. The perplexity results don't necessarily mean that the model can make correct choices for the samples in MMLU, and has less guidance for the model's downstream performance.
5. Missing citations: The authors should also cite [1] for the normalization after retention, and discuss the details of the triton implementation of RetNet and its difference from the implementation in the Flash Linear Attention [2] library.

[1] Zhen Qin, Xiaodong Han, Weixuan Sun, Dongxu Li, Lingpeng Kong, Nick Barnes, and Yiran Zhong. The devil in linear transformer. In Proceedings of the 2022 Conference on Empirical Methods in Natural Language Processing, pages 7025–7041, Abu Dhabi, United Arab Emirates, Dec. 2022. Association for Computational Linguistics.

[2] Yang, Songlin and Zhang, Yu. FLA: A Triton-Based Library for Hardware-Efficient Implementations of Linear Attention Mechanism. https://github.com/sustcsonglin/flash-linear-attention

**Questions:**

Can you also add the results on MMLU and Qasper with the standard metrics besides perplexity?

**Limitations:**

No, the authors should have a limitation section to point out the strong assumptions of their approximation of self-attention and relative position embedding.

---

> ### Author Rebuttal · Authors · 2024-08-07
>
> >Q1: The authors introduce a new term called "Retention," but this is essentially the same as Linear Attention without the denominator, which has already been proposed in `The devil in linear transformer`.
>
> A1: The term is proposed to avoid confusion with the pioneer work "Transformers are RNNs: Fast Autoregressive Transformers with Linear Attention", where linear attention is with the denominator. We can add the discussion of `The devil in linear transformer` (Transnormer) in the paper. We also list several key differences as follows.
>
> - We derive the design starting from the recurrent view, rather than empericially modifying the original attention.
>
> - The theoretical derivations naturally have position embedding and the decay term ($\gamma$), which is critical for stable training and good performance. As shown in Table 3, the ablation "without $\gamma$ decay" performs worse than the proposed retention. Directly using the QKV implementation can not easily converge as expected at larger scale. The Transnormer model additionally interleaves  diagonal blocked sparse softmax attention.
>
> - The three equivalent computation paradigms, i.e., parallel, recurrent, and chunkwise recurrent representations, are also important features of retention.
>
> - Although the overall forms seem similar, but the specific equations are still different. More importantly, the modeling and training behaviors are very different, based on the theoretical nature of the proposed method.
>
> We appreciate that you point out this issue. We are open to discuss these key elements and discuss Transnormer in the paper.
>
>
> ---
> >Q2: Lack of comparison with the baselines on open source pretraining data. All the training experiments are conducted on in-house data mixtures, which harms the reproducibility.
>
> A2: The training corpora used in the paper are all public available. They can be easily downloaded online for acadamic purpose. We use the same training data across models for fair comparisons.
>
>
> ---
> >Q3: The paper doesn't compare RetNet with other linear attention model (such as GLA, RWKV, Mamba) on downstream tasks with standard metrics instead of perplexity. Table 2 only include RetNet and Transformer.
>
> A3:
>
> - As shown in Table 1, we compare RetNet with Hyena, RWKV, Mamba, H3 on fine-grained language modeling evaluation and MMLU answer perplexity. The fine-grained language modeling evaluation correlates well with downstream tasks.
>
> - As pointed out in [1][2], robustly evaluating accuracy metrics need larger model size. Otherwise the accuracy metrics tend to be affected by the "emergence" issue. Transformer is still a quite strong architecture, so we compare with Transformer with 6.7B model size in Table 2, rather than scaling up all the variants to 6.7B for robust comparisons. The metric protocal is by design for more scientic evaluation.
>
> We hope the above explanation clarifies the rationale behind our experiment designs.
>
> [1] Are Emergent Abilities of Large Language Models a Mirage?
>
> [2] Understanding Emergent Abilities of Language Models from the Loss Perspective
>
>
> ---
> >Q4: The efficiency measurment of RetNet+ is absent.
>
> A4: The efficiency measurment of RetNet+ can be obtained by interpolating the efficiency results of hybrid blocks, depending on the ratio of the mixed blocks, which are provided in the main content.
>
>
> ---
> >Q5: The evaluation on MMLU/Qasper is using perplexity but not the widely-used accuracy/F1 metric. The perplexity results don't necessarily mean that the model can make correct choices for the samples in MMLU, and has less guidance for the model's downstream performance.
>
> A5: The metric protocal is by design for more scientic evaluation. As pointed out in [1][2], robustly evaluating accuracy metrics need larger model size. Otherwise the accuracy metrics tend to be affected by the "emergence" issue. In comparison, we report accuracy numbers for scaling-up settings, such as Table 2 (i.e., scaling up size) and Table 5 (i.e., scaling up training tokens and size).
>
> [1] Are Emergent Abilities of Large Language Models a Mirage?
>
> [2] Understanding Emergent Abilities of Language Models from the Loss Perspective
>
>
> ---
> >Q6: Missing citations: The authors should also cite [1] for the normalization after retention, and discuss the details of the triton implementation of RetNet and its difference from the implementation in the Flash Linear Attention [2] library.
>
> A6: We can add them in the camera-ready version.

---

### Official Review · Reviewer_EVGT · 2024-07-15

**Soundness:** 3
**Presentation:** 3
**Contribution:** 3
**Rating:** 6
**Confidence:** 4

**Summary:**

This paper presents Retentive Network (RetNet), a family of efficient models that incorporate exponential decay within a linear attention-like structure. RetNet shares similarities with state-space models and linearized attention, enabling both training parallelism and O(1) inference cost. Additionally, RetNet supports chunk-wise parallel computation for efficient long-sequence training. Experimental results demonstrate RetNet achieves performance comparable to Transformers and outperforms other efficient variants on language modeling and vision tasks.

**Strengths:**

- The structure of RetNet is easy to understand and follow
- RetNet exhibits promising training and inference efficiency, and is able to scale up to 6B.
- Comprehensive evaluation on both language and vision tasks, highlighting its generalizability.

**Weaknesses:**

- Some experiments could be improved
- Some claims may be misleading
- RetNet's performance lags behind Transformers at smaller model scales, suggesting it might be more demanding in terms of capacity and compute resources for optimal performance. This trade-off should be carefully considered and analyzed.

**Questions:**

1. The inference results in Figure 5 start with 2048. What’s the inference speed for shorter sequences?
2. The claim that "None of the previous work can achieve strong performance and efficient inference at the same time compared with Transformers" is overly strong and potentially misleading. Recent advancements in efficient modeling, such as Mamba, have demonstrated better scaling properties than Transformers.
3. It would be great to include the training loss curve for both Transformer and RetNet.

**Limitations:**

I didn't see serious problems.

---

> ### Author Rebuttal · Authors · 2024-08-07
>
> Thank you for the positive review.
>
> >Q1: The inference results in Figure 5 start with 2048. What's the inference speed for shorter sequences?
>
> A1: The RetNet inference speed remains almost constant across length, and Transformers' speed can be extrapolated according to Fig 5. As shown in Figure 5(c), we also compare inference latency of Transformer with 1024 length (yellow). It shows there is still improvement for sequences shorter than 2048.
>
>
> ---
> >Q2: The claim that "None of the previous work can achieve strong performance and efficient inference at the same time compared with Transformers" is overly strong and potentially misleading. Recent advancements in efficient modeling, such as Mamba, have demonstrated better scaling properties than Transformers.
>
> A2: We can improve this part as suggested to take more recent advances into consideration.
>
>
> ---
> >Q3: It would be great to include the training loss curve for both Transformer and RetNet.
>
> A3: We saved the loss curves in Tensorboard. We can provide them for future research work.

---

### Author Response · Authors · 2024-08-12

Dear Reviewer,

Thank you for taking the time to review our submission.

We have carefully considered your feedback and submitted our responses and clarifications during the rebuttal phase. We highly value your insights and would appreciate any further discussion or questions you might have regarding our responses.

We understand that you have a busy schedule, but your additional feedback would be incredibly helpful for us to improve our work. If you could take a moment to review our rebuttal and provide any further comments, we would be very grateful.

Thank you again for your time and consideration.

Best regards

---

### Comment · Senior_Area_Chairs · 2024-08-13

Dear reviewers,

The discussion period will end soon. If you haven't responded to the authors' rebuttal, please do so and kick off the discussion.

Best,
SACa

---

### Decision · Program_Chairs · 2024-09-25

**Decision:**

Reject

**Comment:**

The paper proposes the Retentive Network, which is a kind of linear attention network. All reviewers agree that the idea (if novel) is interesting.

However, Reviewer maEf pointed out that the proposed approach highly resembles an EMNLP'22 paper [1]. In the author response, the author emphasized different roots of their design and some minor difference. However, this does not seem to warrant a main conference paper. Also, there lacks empirical comparison between the two models.

The other two reviewers give acceptance-side scores (borderline acc & weak acc), but their original reviews did not factor in the EMNLP'22 paper. In the reviewer-AC discussion, the reviewers read the related paper and acknowledged the resemblance.

The authors should honestly discuss and compare with [1] in their next version, and more importantly, reconsider the significance of their contributions.

[1] Zhen Qin, Xiaodong Han, Weixuan Sun, Dongxu Li, Lingpeng Kong, Nick Barnes, and Yiran Zhong. The devil in linear transformer. In Proceedings of the 2022 Conference on Empirical Methods in Natural Language Processing, pages 7025–7041. 2022.